# Mid-Holocene extreme precipitation in the Tibesti, Central Sahara

Philipp Hoelzmann [1,7] ✉, Martin Claussen [2,3,7] ✉, Anne Dallmeyer [2], Frank Darius [1], Michèle Dinies [1], Christian Reinhardt-Imjela [1], Leonore Jungandreas [4], Birgit Schröder [5] & Stefan Kröpelin [6]

During the early and mid Holocene deep lakes existed in the Tibesti volcanic complex, the highest mountain in the Sahara, but it is still unclear why they formed. Here, we combine sedimentary data analyses, high-resolution climate modelling and water balance modelling to provide a quantitative and differentiated estimate of the mid-Holocene water balance of the Trou au Natron (Doon Orei) and Era Kohor crater lakes and to discuss the underlying atmospheric circulation. During the mid-Holocene, the Tibesti received at least an order of magnitude more precipitation than the surrounding plains due to strong orographic uplift of moist air masses, which were surprisingly brought in by north-easterly winds from the Mediterranean, not by the stronger West African monsoon from the south. This may explain differences in the water levels between the Trou au Natron in the north-western part of the Tibesti and the Era Kohor in the south-eastern, leeward part. Our analysis demonstrates the importance of orographic precipitation for evaluating the hydroclimate of the central part of the Sahara – a factor grossly underestimated in the global climate models commonly used for palaeo and future climate simulations.

The precession-forced intensification of the summer monsoon during the African Holocene Humid Period (AHHP) is well documented by multiple climate archives across North Africa[1–8], while data from Saharan mountain ranges are scarce. Remnants of lacustrine sediments in the wide, steep and nowadays dry craters of the Tibesti – the Sahara's largest and highest mountain range – witness to the past existence of high-altitude deep lakes. This extremely remote and insecure[9] region has been discovered in 1869[10], explored in 1915[11] and studied by geoscientists since the 1960s[12–18], but only recently plant remains and oxygen isotopes from diatoms of palaeolake deposits in the Trou au Natron (Doon Orei – big hole in the local Teda dialect) and Era Kohor craters were newly dated and investigated to put these into the context of the African Holocene Humid Period (AHHP)[19,20].

The Trou au Natron crater covers some 40 km² and is located at the foot of the 3296 m high Pic Toussidé in the north-western part of the Tibesti at 20°58′N, 16°33′E. The ten times smaller Era Kohor sub-crater covers only 4.1 km² and is located at the south-eastern edge of the Tibesti at 19°50′N, 18°33′E, inside the large Emi Koussi summit caldera (84 km²). With 3415 m above sea level, it is the highest mountain in the Sahara. The craters were apparently filled by the deepest endorheic lakes of the entire Sahara with possible water depths of more than 330 m and 130 m, respectively (Fig. 1).

The development of the lakes is considered to be broadly synchronous[20] and is assumed to be linked to changes in the West African monsoon dynamic as its intensification has provided wetter conditions across much of Africa during the AHHP. The lakes

¹Institute of Geographical Sciences, Freie Universität Berlin, Berlin, Germany. ²Max-Planck-Institut für Meteorologie, Bundesstr. 53, Hamburg, Germany. ³Center for Earth System Research and Sustainability (CEN), Universität Hamburg, Bundesstr. 53, Hamburg, Germany. ⁴German Centre for Integrative Biodiversity Research (iDiv), Leipzig, Germany, now at: Institute for Earth System Science and Remote Sensing, Universität Leipzig, Leipzig, Germany. ⁵GFZ Helmholtz Centre for Geosciences, Telegrafenberg, Potsdam, Germany. ⁶Institute of Prehistoric Archaeology - African Archaeology, Universität zu Köln, Universität zu Köln, Köln, Germany. ⁷These authors contributed equally: Philipp Hoelzmann, Martin Claussen. ✉e-mail: philipp.hoelzmann@fu-berlin.de; martin.claussen@mpimet.mpg.de

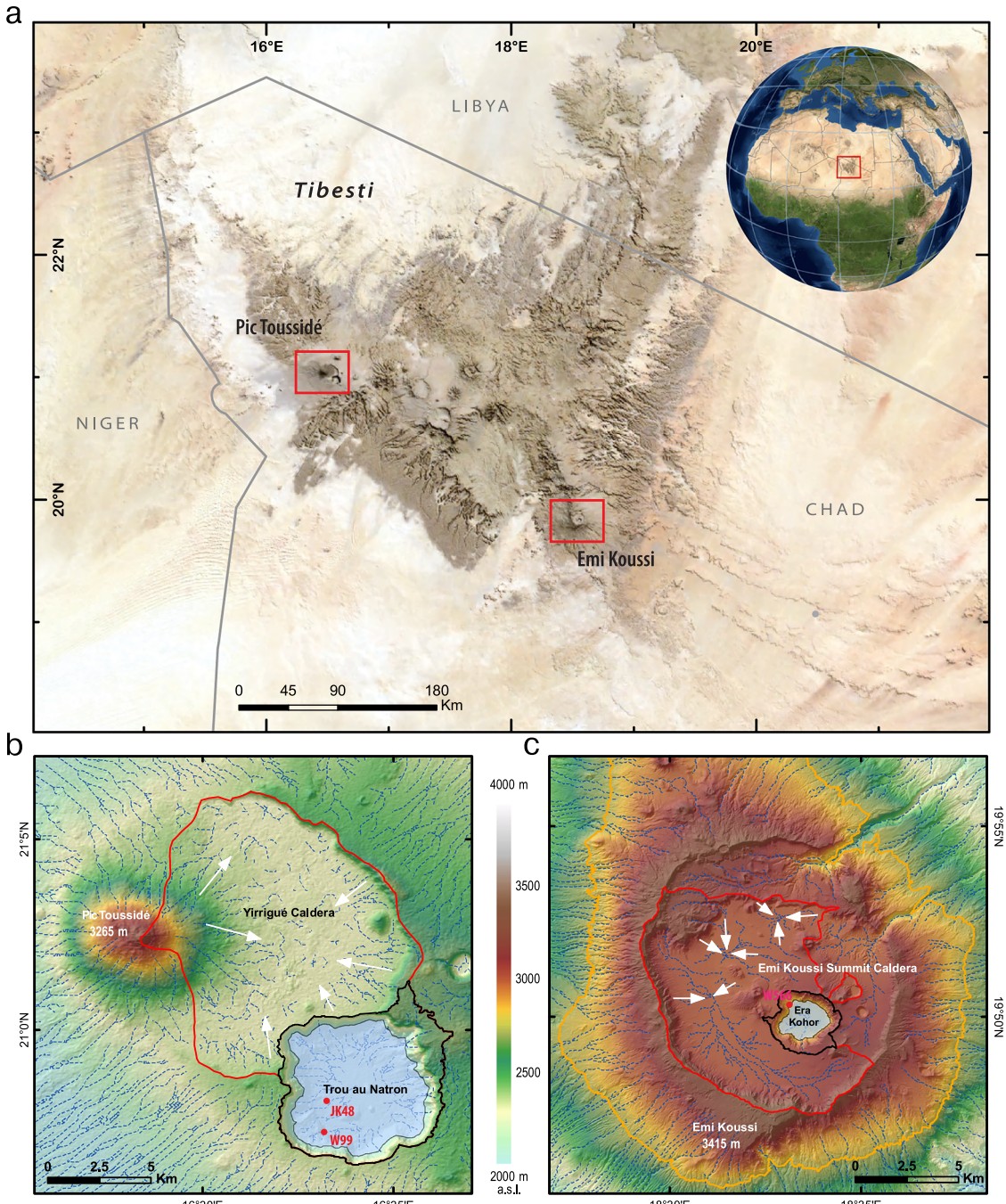

**Fig. 1 | Satellite image of the Tibesti Mountains and close-up maps of the Trou au Natron and Era Kohor craters. a** Tibesti Mountains within the Central Sahara. **b, c** Contour-level maps, drainage systems and catchments of the Trou au Natron (TaN) (**b**) and Era Kohor (EK) (**c**) craters. White arrows show the direction of endorheic drainage and red lines define the endorheic catchment in the craters. Black lines delimit the surface catchments of the lakes with TaN 17 km² and EK 4.4 km², respectively. The reconstructed maximum lake areas within both craters are shown by the light blue colour: 31.6 km² for the TaN and 3.1 km² for the EK, respectively. Data sources for **a** MODIS/BMNG (Stöckli et al.[63] and Natural Earth (naturalearthdata.com); **b, c**[64] Sentinel-2 Mosaic data and[65] JAXA ALOS DSM World 3D-30 m (eorc.jaxa.jp); all publicly available.

experienced wet conditions before 9.7 cal ky BP, the earliest date of the records. Gradual, progressive drying started around 6.5 cal ky BP and intensified after 5.3 cal ky BP[19,20]. However, with the relatively coarse temporal resolution used in Yacoub et al.[20], only general differences between the lakes were identified. Likewise, a quantitative estimate of the atmospheric moisture transport, of the water budget requirements of these lakes and the identification of the cause of large hydrological differences between the lakes has not been undertaken.

Palaeoclimate modelling has not been able to shed light on this either: Earth System Models (ESMs) struggle to simulate the Holocene monsoon expansion into the Sahara and fail to capture early to mid-Holocene precipitation patterns[21], likely due to missing feedback processes[22]. We suppose that steep orography could have a strong effect on the atmospheric circulation and, hence, on precipitation in the Holocene Sahara – a process which has received little attention so far. ESMs can currently be employed at a rather coarse spatial resolution for long-term palaeoclimate simulations in which steep orography

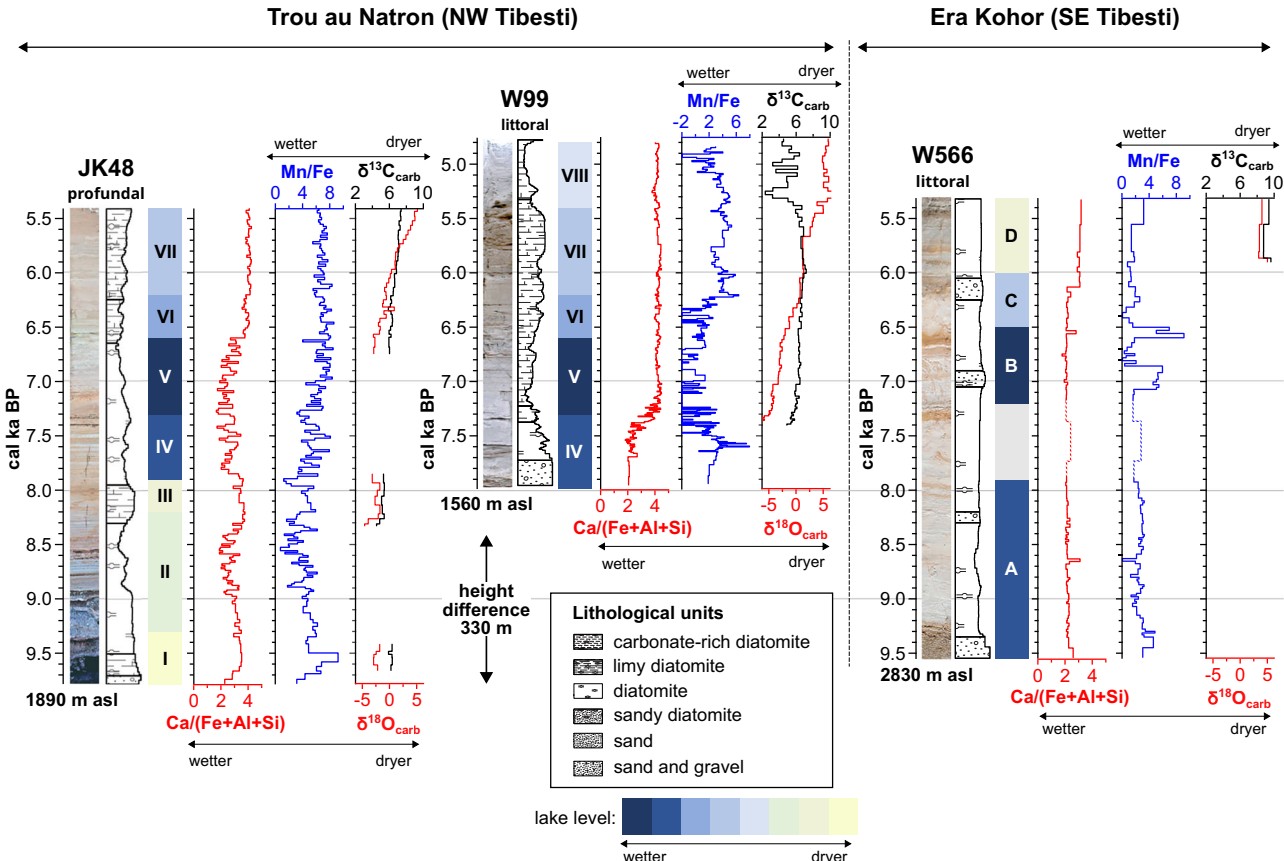

**Fig. 2 | Crater palaeolake records of the Trou au Natron (JK48 and W99) and Era Kohor (W566).** Stratigraphy, lithology, selected geochemical data and identified lake phases for the crater palaeolake records. Centered log-ratio (clr) Ca/(Fe+Al+Si) ratios are related to increasing authigenic carbonate precipitation under dryer conditions. Lower clr Mn/Fe ratios are interpreted as signs for anoxic conditions due to an increasing lake level. The $\delta^{18}O_{carb}$ values show a broad range from −4‰ in the lower part to +5‰ in the upper part mainly reflecting changes in the hydrological balance between precipitation $P$ and evaporation $E$. Generally high $\delta^{13}C_{carb}$ values in all profiles indicate high productivity by diatoms and show a slight increase with evaporation. Decreasing $\delta^{13}C_{carb}$ values at the upper part of W99 indicate decay of organic matter due to a lower lake level.

is severely smoothed. Thus, the height and orographic details are grossly underestimated in ESMs, making local hydrological assessments in mountainous regions highly unreliable.

We aim to identify the moisture sources and palaeoclimate conditions necessary to fill the craters and maintain the palaeolakes in three ways. First, we re-assess the sedimentary records from the Trou au Natron and the Era Kohor described in Yacoub et al.[20], by using complementary sedimentological-geochemical analyses with a much finer temporal resolution, which allows us to study the lake dynamics more accurately and to identify differences in their development. Second, we analyse the first regional palaeoclimate simulations of the Sahara and Sahel region with a spatial resolution of some 5 km for a time slice around 7 ky BP that resolve steep orography. Third, we conduct a comprehensive remote sensing and GIS-based terrain analysis of both craters to assess the system's hydrography (palaeolake areas, catchment, drainage systems) and to develop a simple mathematical model of the equilibrium water balance of the mountain palaeolakes based on the output from the regional palaeoclimate simulations.

## Results and Discussion
### Sediment analyses of the crater palaeolakes
We compare the bulk geochemical, mineralogical and isotopic composition of sandy to carbonate-rich diatomites from the northwestern (Trou au Natron) and southeastern (Emi Koussi) parts of the Tibesti to focus on a detailed interpretation from a sedimentary perspective. Two lake sediment sequences from the Trou au Natron crater were

studied: the lower, profundal section JK48 (1560 m asl) near the crater floor covers the period from c. 9.7 to 5.5 cal ky BP, while the littoral section W99 (1890 m asl) from the southern crater slope covers the period from 7.9 to 4.8 cal ky BP (Fig. 2). The time resolution of the sections is on average 18 years for JK48 and 11 years for W99, i.e. more than an order of magnitude finer than in Yacoub et al.[20]. Both sections show a timely overlap between 7.9 and 5.4 cal ky BP when lake volume reached ca. 7.1 km³ with approximately 330 m water depth and a lake surface of 31.6 km².

We identified eight different lake phases (Methods and Supplementary Notes) that begin with initial lake phases (I and II: ca. 9.7 to 8.3 cal ky BP; Fig. 2) only represented in the lower section JK48 near the crater floor, as the lake level has not yet reached the higher positioned location W99. Calcite precipitation and increased $\delta^{18}O_{carb}$ values point to lake level lowering and drier conditions during a period (III: 8.3–7.9 ky cal BP) when the 8.2 ka Northern Hemisphere cooling event[23] also affected North Africa and climatic feedbacks may have even prolonged its duration in the Sahara[24]. However, the age-depth model has not the necessary precision to reliably fix this phase to the 8.2 ka event or to calculate its duration from this section. The highest lake levels (phases IV and V) occurred between 7.9 and 6.6 cal ky BP when sedimentation began also in the higher section W99. With phase VII between 6.6. and 5.4 cal ky BP significantly increasing $\delta^{18}O_{carb}$ values from −2‰ up to +5‰ indicate in both sections a drying trend. The youngest documented lake phase (VIII; 5.4 to 4.8 cal ky BP) is reflected solely in the upper section W99 and is characterized by a trend toward progressively drier conditions indicated by continuously

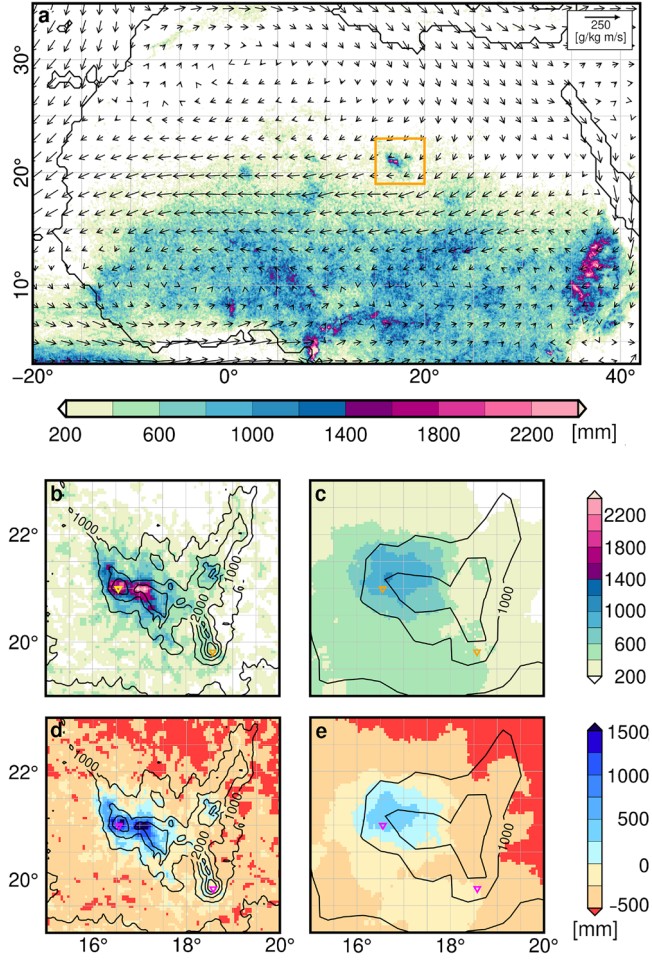

**Fig. 3 | Simulated precipitation and moisture fluxes. a** Simulated precipitation sum (coloured shades) and vertically integrated atmospheric moisture flux (vectors) for the months June to October for a strong monsoon year at around 7 ka BP in the 5 km simulation. Zoom on the Tibesti region (yellow square in a) for precipitation (**b, c**) and the difference of precipitation and potential evaporation (**d, e**). **b,d** depict the results of the high-resolution simulation with grid-box size of 5 km, (**c**) and (**e**) refer to the results of the low-resolution 40 km simulation. Thin lines show contour lines in elevation with 500 m intervals. The Trou au Natron is located at 20.993°N, 15.552°E, and the Era Kohor at 19.817°N, 18.574°E in the model grid. The craters are also indicated by yellow (in **b**, **c**) and magenta (in **d**, **e**) triangles.

increasing water salinities and calcite precipitation with lower $\delta^{13}C_{carb}$ probably due to a decreasing lake level and decay of organic matter. However, a final lake phase after 4.8 cal ky BP is not represented in either of the sections due to erosion.

Section W566 in the Era Kohor crater within the summit caldera of Emi Koussi is located 130 m above the present crater floor. It consists of sandy diatomite and represents a littoral section of the palaeolake which occupied the crater between 9.5 and 5.4 cal ky BP[19,20]. The topographic position of section W566 indicates lake formation prior to 9.5 ky cal BP. Four lake phases were distinguished mainly from the interpretation of Mn/Fe. The general trend of the lake phases is similar to the Trou au Natron, but not identical. The onset of the lake started earlier in the Era Kohor than in the Trou au Natron, with the lake reaching a depth of 130 m by 9.5 cal ky BP (phase A from 9.5 to 7.9 cal ky BP). The highest lake level (phase B; from 7.9 to 6.5 cal ky BP) followed with a maximum lake volume of ca. 0.45 km³ – only 4.3% of the Trou au Natron palaeolake volume. Thereafter phase C represents a lower lake level before authigenic calcite production started around 6.2 cal ky BP (phase D; corresponding to lake phase VII in the Trou au

Natron) reflecting further lake level lowering at least until 5.4 cal ky BP. The top of section W566 is also truncated by erosion.

Both locations show early to mid-Holocene lake evolution congruent to the AHHP[19]. The Trou au Natron crater palaeolake is fed by water that must result from a substantial increase in precipitation, as shown by significantly low $\delta^{18}O_{carb}$ values of −6‰ at 7.2 cal ky BP reflecting lake water mainly from precipitation influenced by the altitude and/or the amount effect[25]. The continuously increasing $\delta^{18}O_{carb}$ values in positive covariation with $\delta^{13}C_{carb}$ indicate the mid-Holocene drying trend of a decreasing ratio of precipitation to evaporation towards the end of the AHHP. The records are palaeoclimatologically significant due to the dependence on precipitation-fed groundwater, also reflected in the Trou au Natron by the distinct short-term lake level lowering (Fig. 2) possibly related to the 8.2 ky Northern Hemisphere cooling event[23]. The deepest lake phases occur thereafter, congruent to the maximum of water body extension in the North African plains, highlighting the role of groundwater recharge[7].

Based on this detailed account of the Holocene lake development, several research questions arise: (1) What were the hydrological water balances for the palaeolakes? (2) Which atmospheric circulation patterns favour the production of enough precipitation for the existence of the crater lakes? (3) Why do the estimated lake volumes for the deep lake phases differ so strongly between the Trou au Natron and the Era Kohor?

## Simulated precipitation and atmospheric moisture flux

To estimate precipitation and large-scale moisture flux over the Tibesti during the mid-Holocene, we analyse two high-resolution regional climate simulations of the monsoon season (June-October) at around 7 ky BP. The first simulation represents conditions of a strong mid-Holocene monsoon season, while the second represents a weak mid-Holocene monsoon season[26] (Methods). The entire Tibesti stands out with precipitation levels much higher than those in the surroundings (Fig. 3). In the upscaled annual mean (Methods), the grid cell representing the Trou au Natron receives some 1960 mm annual precipitation during the strong monsoon year, and the grid cell corresponding to the Era Kohor receives about 530 mm. During a weak monsoon year, precipitation reduces to some 830 mm and 230 mm for the Trou au Natron and the Era Kohor, respectively (Supplementary Table 2, 3).

In these simulations, particularly northerly and easterly winds related to the African Easterly Jet (characterized by blue colours in Fig. 4b, Fig. 5a, e), which was displaced northward during the mid-Holocene compared to its location today, carry most of the moisture to the Tibesti on vertical average over the troposphere. It may seem surprising that the Mediterranean Sea is the main moisture source for precipitation over the Tibesti, as it was previously assumed that a stronger southerly West African monsoon could have been the main factor[20]. However, it is not moisture availability itself which produces precipitation, but the convergence of the moisture flux and vertical uplift of air masses. Thus, little precipitation is expected over the rather flat Calanshio Sand Sea (Libyan Desert) north and upstream of the Tibesti. But the strong uplift of air at the steep Tibesti Mountains and the regional atmospheric conditions favoured precipitation: During August, the rainiest month in this region, the southern slopes are embedded in the low-level monsoon flow (characterized by the red colour in Fig. 4a, Fig. 5b, f), which was much stronger during the mid-Holocene than today[22,27]. The south-westerly monsoon winds transport moisture from the Atlantic to the southern and western side of the Tibesti (Fig. 4a), but the monsoon layer is shallow and does not reach up to the lakes (Fig. 4a, b, Fig. 5b, f). Instead, northeast winds carry moisture from the Mediterranean Sea to the northern slopes, where the wind converges and air is lifted (Fig. 5b, f). The upward motion is particularly strong above the Trou au Natron region, as the northern

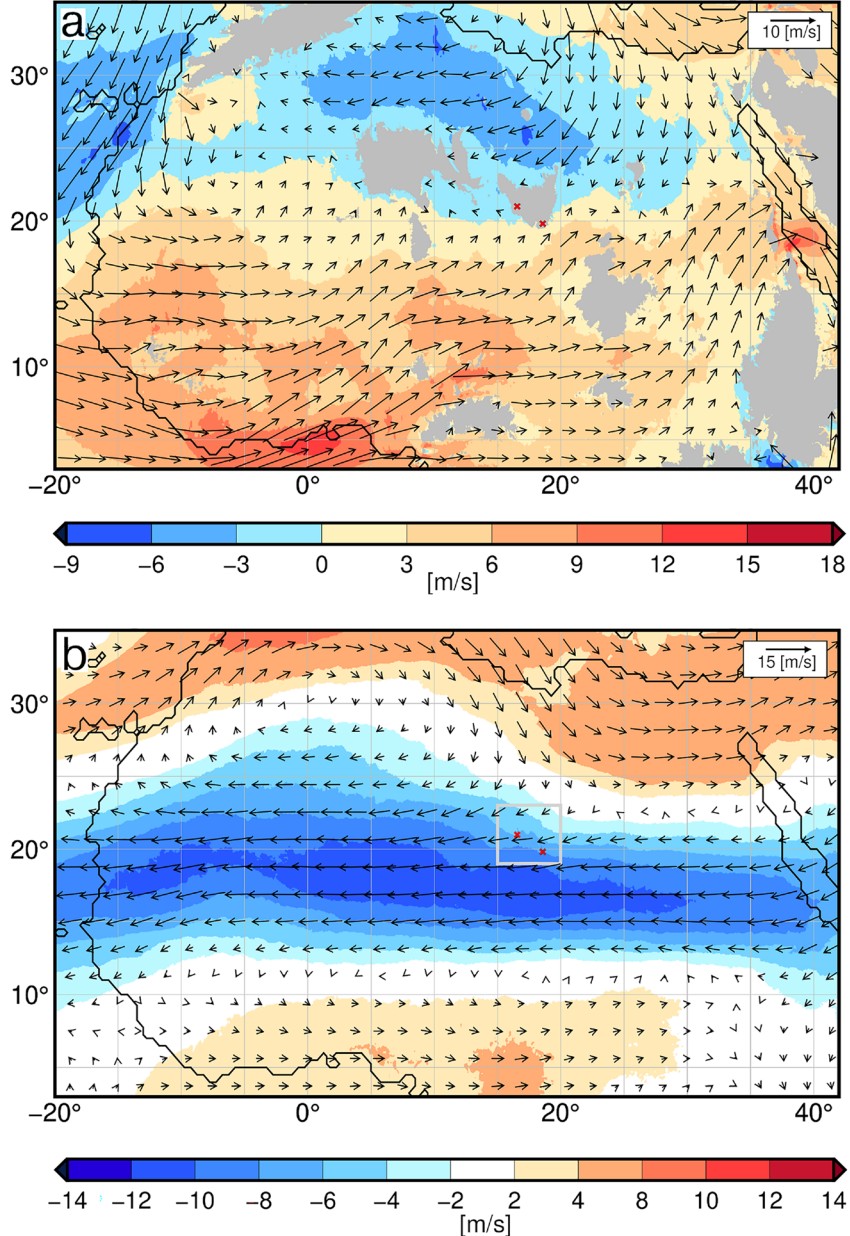

**Fig. 4 | Simulated horizontal wind over northern Africa. a** near the ground at 925hPa (in the monsoon layer) and **b**, at the core level of the African Easterly Jet at a height of 600hPa (approximately 4 km) at around 7 ky BP for the month of August in a strong monsoon year. The vectors depict the strength and direction of the horizontal wind. Blue colour shades indicate the strength of the easterly wind component, and red colour shades, the westerly wind component. (The blue colour in **b** show the African Easterly Jet.) The little red crosses show the location of the palaeo lakes of the Trou au Natron in the northern part of the Tibesti and the Era Kohor in the southern part. Grey shades in the upper figure indicate orography above 925 hPa.

part of the Tibesti is oriented in a west-east direction. The upward motion reaches through the entire troposphere, in levels even below freezing point, which explains the high precipitation rates (Fig. 5c, d).

The Era Kohor is located inside the Emi Koussi at the southern edge of the north-south oriented branch of the Tibesti. Therefore, northerly winds are slowed down and rained out on their way over the mountainous terrain before reaching the lake. The upward motion above the Emi Koussi region is smaller than over the Trou au Natron region (Fig. 5). During August, the monsoon air from the south (Fig. 4a) flows up the steep mountains below the Emi Koussi (Fig. 5f), but the uplift is too weak to reach the lake. The mid-level easterlies are also only slightly lifted. In addition, the African Easterly Jet accelerates strongly downstream of the Emi Koussi, thereby transporting moisture away from the lake (Fig. 4b, Fig. 5e). This flow configuration leads to

smaller precipitation rates in the Emi Koussi region (Fig. 5g, h) than in the Trou au Natron region (Fig. 5c, d).

The effect of steep orography on the amount of simulated precipitation is clearly seen when comparing precipitation and potential evaporation between the simulations at 5 km grid and at 40 km grid (Fig. 3b–e). The orography is much smoother and much less detailed in the 40 km simulation (Fig. 6) and surely in a global simulation which run at a horizontal resolution of some 200 km and more. Consistently, the precipitation in the Tibesti region is much smaller in the 40 km simulation than in the 5 km simulation (Fig. 3b, c) and the extreme local precipitation maxima seen in the 5 km simulation are absent in the 40 km simulation. In contrast the potential evaporation does not change very much such that the net precipitation (precipitation minus evaporation) substantially decreases and becomes strongly negative

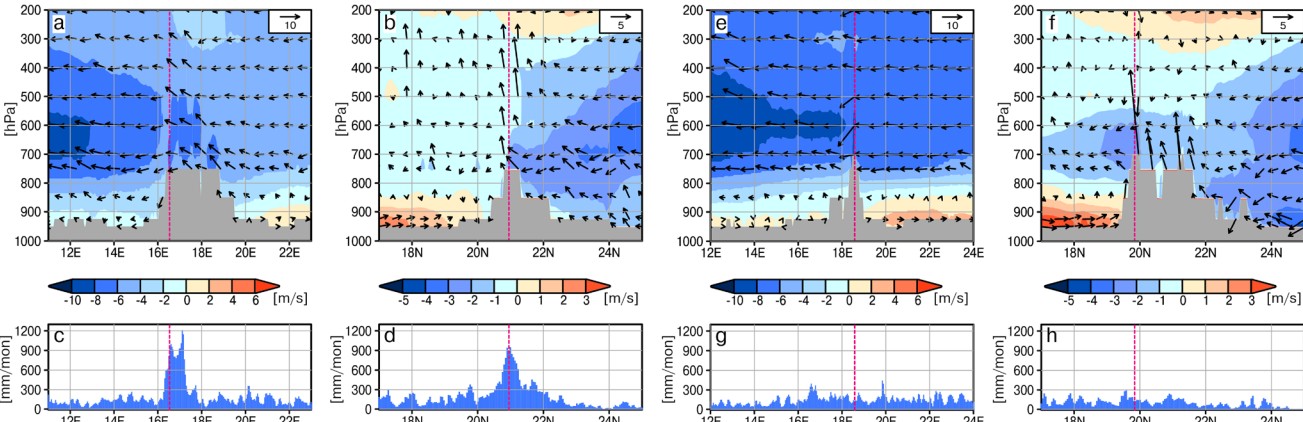

**Fig. 5 | Simulated mid-Holocene atmospheric circulation and precipitation at the crater lakes Trou au Natron and Era Kohor during August.** Shown are the simulated zonal and meridional wind components along the latitude 20.993 N (**a**) and the longitude 16.552E (**b**) cutting through the Tibesti at the location of the Trou au Natron in the model and along the latitude 18.574 N (**e**) and the longitude 19.817 E (**f**) of the Era Kohor. The vectors depict the zonal and vertical wind components in (**a, e**), and the meridional and vertical wind components in (**b, f**). Please note that the vertical component of the wind vectors is amplified by a factor of 100 compared

to the horizontal component. The shades indicate only the strength of the westerly component of the wind in (**a, e**) and the southerly component in (**b,f**). West winds are indicated by positive values and orange-red shading, and the dominating east wind is shown by negatives values and a blue shading in (**a,e**). In the meridional cross sections in (**b,f**), positive values and orange-red shading indicate south winds and negative values and blue shading north winds. The precipitation distributions along the cross sections cutting through the Trou au Natron are displayed in (**c, d**), and for the Era Kohor in (**g, h**).

(Supplementary Table 2) in all years, except for the strong monsoon year at the Trou au Natron (Fig. 3d, e). The Era Kohor is located at a much lower height in the 40 km simulation (Fig. 6h) and, hence, receives more precipitation from the south, i.e., from the monsoon flow. Therefore, the amount of precipitation in the 40 km simulation is almost as large as in the 5 km simulation at the Era Kohor, but for the wrong reason. Because of a larger potential evaporation, the net precipitation at the Era Kohor is, nonetheless, even more negative in the 40 km simulation than in the 5 km simulation (Supplementary Table 3). This comparison between the 40 km and 5 km simulations also underlines the asymmetrical effect of the orography on the precipitation pattern along the Tibesti, favouring precipitation around the Trou Natron while being much less pronounced around the Era Kohor.

**Estimate of the water balance of the crater palaeolakes**
Besides precipitation $P$, it is necessary to calculate the potential evaporation $PE$ from the lake surface, the runoff into the lake and the net groundwater flow over some long-term mean in order to assess whether permanent lakes in the Trou au Natron and the Era Kohor could exist in the atmospheric conditions found in our mid-Holocene climate simulation.

The potential evaporation depends on various factors such as the wind and the turbulence over the lake surface and the energy budget at the lake surface. Because these factors are not known in detail, we use Budyko's approach[28], according to which the $PE$ is directly proportional to the net radiation at the lake surface. The variability of the monsoon strength is extrapolated from the regional climate simulation at the horizontal resolution of 40 km to the simulation at the fine scale of 5 km (Methods), resulting in a positive atmospheric water budget ($P - PE$) of some 150 mm/y with a standard deviation of 34% for the Trou au Natron (Fig. 3c, Supplementary Table 2), and a generally negative budget of some −640 mm/y with a standard deviation of 36% for the Era Kohor (Fig. 3c, Supplementary Table 3).

The topographical analysis of the drainage systems in the Tibesti is shown in Fig. 1. The palaeolakes receive water from precipitation on their lake surfaces and can also receive water by groundwater infiltration and as surface runoff from their catchment areas: 17 km² around the Trou au Natron and 4.4 km² around the Era Kohor (excluding lake areas). The rims of both craters rise above the caldera floors preventing surface runoff from other parts of the calderas from

flowing into the lakes. However, because of the endorheic drainage systems of the large calderas, runoff generated outside the lakes' direct catchment accumulates in the deepest part of the caldera floors and infiltration could thus support the development of local groundwater that fed the lakes. A similar drainage system can be observed under recent conditions in Darfur about 1000 km further southeast at the semi-arid Deriba caldera of Jebel Marra with precipitation up to 1000 mm/y (at around 13°N, 24.25°E[29,30]). As no outflows exist at the Trou au Natron or the Era Kohor (closed basins), lake water loss is limited to evaporation from the open water surface and leakage through the lake bottom.

For the Trou au Natron, climate and hydrographic modelling (Methods, Supplementary Table 2) yields an excess of water from ($P - PE$) plus runoff $R$ of some 580 mm/y with a standard deviation of 80%. For the Era Kohor, the water budget ($P - PE + R$) is negative and amounts to almost −300 mm/y (+/− 81%) (Supplementary Table 3). This result implies that in the Trou au Natron, a stable lake could exist at around 7 ky BP and that water likely leaked from the lake into the ground. For the Era Kohor, however, a percolation of groundwater into the lake should have kept the crater lake stable. A percolation of water into the Era Kohor is plausible from the assessment of the drainage system of the large Emi Koussi caldera within which the Era Kohor sub-caldera is embedded. These results qualitatively agree with the estimates of Yacoub et al.[20], who found a ratio $PE/(P + R) < 1$ for the Trou au Natron and phases of $PE/(P + R) > 1$ for the Era Kohor for the centuries around 7 ky BP.

**Substantial moisture supply from the Mediterranean**
By combining proxy data, remote sensing, water balance estimates and high-resolution palaeoclimate simulations, we were able to understand why the regional water budget during the AHHP was high enough to sustain lakes in the Tibesti, despite of their location in the Central Sahara. The Trou au Natron palaeolake experienced the highest lake level during mid-Holocene (around 7 ky BP), while the Era Kohor palaeolake was already characterized by high lake levels during the early Holocene (around 9.5 ky BP). What could have been the reason? During the mid-Holocene, the Mediterranean Sea was the main source of moisture for the Tibesti mountains, whose steep slopes more efficiently received precipitation than the flat surroundings due to the vertical uplift. The West African monsoon reached the Tibesti from the

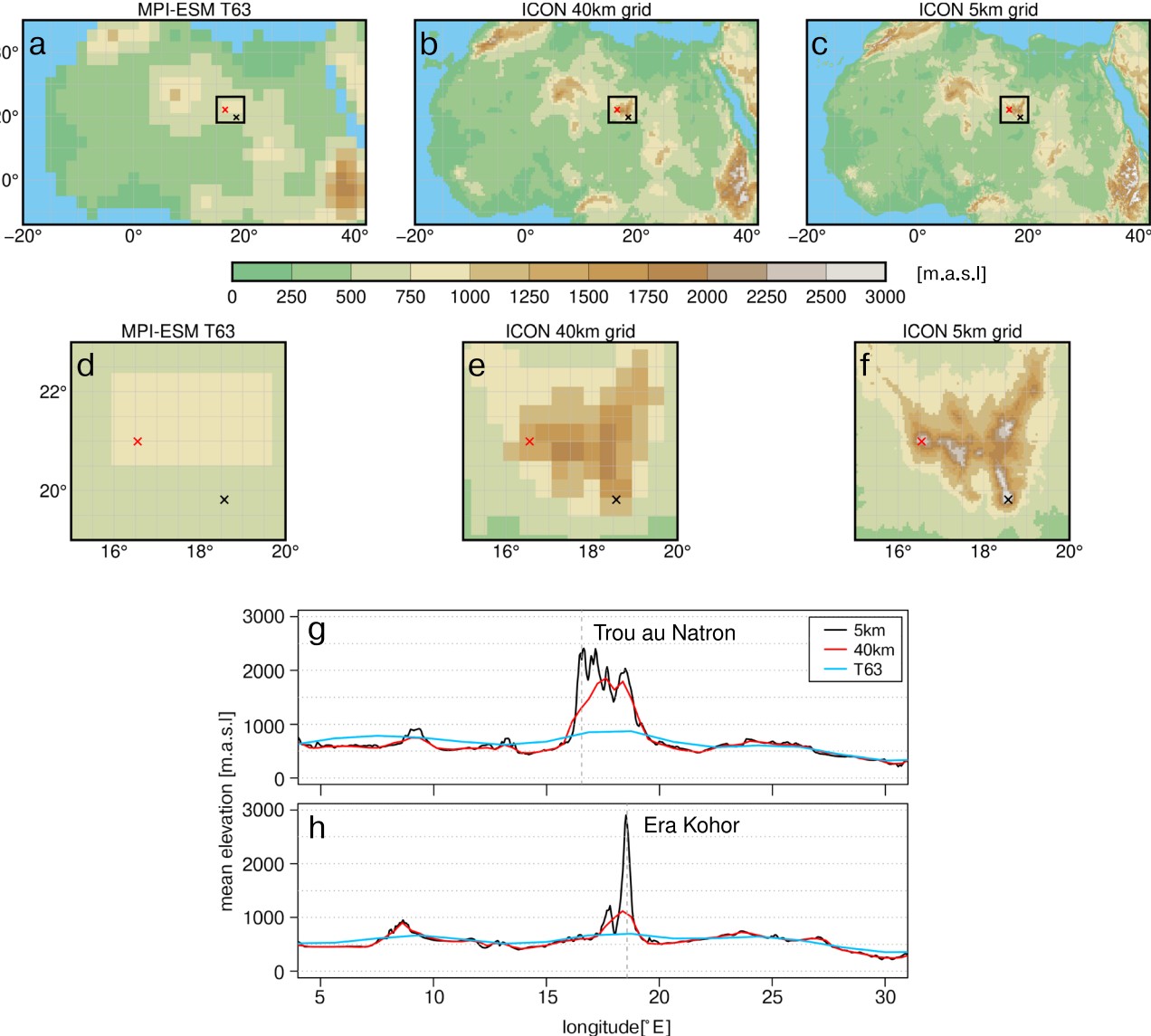

**Fig. 6 | Orography of northern Africa seen in models of different resolution.**
**a** Mean orography in the global model MPI-ESM (T63 grid: ~200*200 km), which provides the initial and boundary conditions for the regional simulation; (**b**) in the ICON-NWP 40 km resolved simulation and **c** in the ICON-NWP 5 km resolved simulation. Centre panel show a zoom into the Tibesti region (**d**) for the T63 resolution, **e** for the 40 km and **f** for the 5 km resolution. Panel (**g,h**) show the orographic heights for latitudinal cross sections through the Trou au Natron (red cross) and Era Kohor (black cross) in T63 (blue), 40 km (red) and 5 km (black) resolution. Heights are given in meter above sea level [m.a.s.l].

south, but it was too weak to be lifted up to the high crater lakes during that time. For the early Holocene, however, the monsoon flow was likely more intense[31–34] implying that the monsoon may have made a substantial contribution to moisture supply during that time. Hence the Trou au Natron on the east-west orientated branch of the Tibesti should have benefited more from the uplift of the mid-Holocene northerly flow, and the Era Kohor on the southern edge of the north-south orientated branch (see Fig. 3), more from the intense early Holocene southwesterly monsoon flow. In addition, the African East-erly Jet was presumably located further to the north during the earlier Holocene than during the mid-Holocene. This could have led to a stronger moisture transport away from the Trou au Natron and less uplift contributing to the temporal difference in the water budgets of the crater lakes between early and mid-Holocene manifested in the proxy records.

Our results not only constrain palaeohydrological changes between the Saharan mountains and plains during the AHHP much more accurately but can also improve assessments of scenarios of potential future changes. Scenario simulations indicate a wetter and greener Sahel and Southern Sahara in the future as a result of higher $CO_2$ levels[35,36]. However, all studies using coarse-scale climate models fail to capture the effect of the steep slopes of Tibesti on the atmospheric circulation. Thus, these simulations likely underestimate the potential future water sources of the Sahara, but also the risk of extreme hydrological events and their consequences for mountainous terrain.

## Methods

### Sampling of the palaeolake sequences
Within the craters Era Kohor (SE Tibesti, section W566) and Trou au Natron (NW Tibesti, sections W99 and JK48), palaeolake remnants are preserved at the slopes and, in the case of Trou au Natron, near the crater floor as well. These diatomaceous sediments exhibit thicknesses of 147 cm (W566), 231 cm (JK48) and 329 cm (W99). They were sampled in subsequent monoliths in the field and shipped to the laboratory. The monoliths from all three sections were sub-sampled at

1–2 cm sample intervals for most of the monoliths. As some monoliths did not reach the laboratory intact, these segments were sampled with a single bulk sample. This resulted in the following sample numbers: - W566 (Era Kohor; outcrops at 19°50.400'N/18°33.100'E at 2796 m asl, 130 m above the caldera floor) from 0-147 cm: 95 samples; - JK48 (Trou au Natron; yardang at 20°58.349'N/16°33.157'E, 20 m above the caldera floor at 1560 m asl) from 0 to 231 cm: 217 samples; - W99 (Trou au Natron; yardang at 20°57,016 N/16°33,110'E, at 1890 m asl, 330 m above section JK48) from 0 to 329 cm: 275 samples. Each sample was divided into three subsamples for geochemical, diatom[20] and pollen analyses[19].

## Chronology

Due to the absence of terrestrial macroscopic plant remains in the sampled sediments and suspected hard-water and reservoir effects, the age-depth models are based on concentrates of microscopic, charred plant particles. A combination of different protocols with seven preparation steps (Supplementary Notes) was applied. The AMS dating was performed at the Poznan Radiocarbon Laboratory (https://radiocarbon.pl/) and at Beta Analytics (https://www.radiocarbon.com/). The radiocarbon ages were calibrated with IntCal20: Northern Hemisphere[37]. Bayesian approaches with implemented outlier analyses were performed with rbacon (version 2.4.3[38].) to establish age-depth models (Supplementary Notes) for the three sediment sequences.

## Sediment geochemistry

All sediment geochemical analyses were performed on homogenized subsamples from monoliths of the three lacustrine sections. Carbon quantification (TC = total carbon; TOC = total organic carbon; TIC = total inorganic carbon) was undertaken using elemental analysers. A LECO TruSpec device was used to determine TC: up to 150 mg of the subsamples is wrapped in tin foil and burned in an oxygen flow at 950 °C. An infrared detector measures the evolved $CO_2$ for TC. TIC is determined with a Woesthoff Carmhograph C16 analyzer by evolving $CO_2$ from the sample (up to 200 mg) during mixing with hot (70 °C) acid ($H_3PO_4$) and subsequent quantification of the evolved $CO_2$ in 20 ml of a 0.05 N NaOH solution by conductivity. As a calibration standard calcite ($CaCO_3$) is used (12.01 ± 0.14%; RSD < 2%). TOC is calculated by subtracting TIC from TC.

An Analyticon NITON XL3t energy-dispersive x-ray fluorescence spectrometer (P-EDXRF) equipped with a CCD-camera and a semiconductor detector was used to analyse the elemental contents of Al, Ca, Ti, Mn and Fe. The powdery sample (~1 g) was placed into plastic cups that were sealed with a mylar foil (0.4 μm). The cups were mounted on the P-EDXRF and measured for 120 s with different filters to detect specific elements. Only measurements that show values larger than four times the 1 sigma errors were considered. Recovery values were checked for quality control with two certified lacustrine sediments (LKSD-2 and LKSD−4[39]). To ensure constant measurement conditions the CRMs were measured at the beginning and the end as well as every tenth sample. X-ray fluorescence (XRF) based analyses of Ca/(Fe + Ti + Al) provide indications of increased authigenic carbonate precipitation indicating drier conditions[40–42], whereas Mn/Fe ratios are interpreted as reflecting the oxygenation of the water column[43,44] with lower ratios indicating increased palaeolake depth.

Qualitative and semi-quantitative investigation of mineralogical compounds was undertaken using X-ray powder diffraction as the carbonate mineralogy reflects water salinities and specific carbonates may influence the stable isotope ($\delta^{18}O_{carb}$ and $\delta^{13}C_{carb}$) composition of bulk carbonates. The powdered sample was placed in the sample holder and analysed with a RIGAKU Miniflex600 at 24 mA/36 kV (Cu kα) from 2° to 70°(2θ) with a goniometer step velocity of 0.5°/min. The software program X-Pert HighScore Version 1.0b by PHILIPS Analytical B.V. was used to identify and semi-quantify the mineral composition.

## Stable oxygen and carbon isotopes on bulk carbonates

Stable oxygen and carbon isotopes in lake carbonates ($\delta^{18}O_{carb}$, $\delta^{13}C_{carb}$) mainly reflect the hydrological balance between precipitation (P) and evaporation (E), but also the influence of organic productivity and organic decay[45–47]. Authigenic carbonates commonly precipitate by calcium carbonate supersaturation in the epilimnion driven by photosynthetic utilization of $CO_2$ and by evaporative release of $CO_2$. The $\delta^{18}O_{carb}$ composition of authigenic lake carbonates mainly reflects the $\delta^{18}O$ of lake water and thus the hydrological balance influenced by precipitation, groundwater, temperature and evaporation[46,47]. Today, the global annual precipitation isostope map by Terzer et al.[48] shows amount weighted $\delta^{18}O_{ANN}$ predictions ranging between −5.9 and −3‰ in the Sahara region, between −2.9 and 0‰ in the Chadian region and between −8.9 and −6‰ in the higher Tibesti Mountains due to the high altitude and the amount effect.

Stable oxygen and carbon isotope measurements on bulk carbonates ($\delta^{18}O_{carb}$, $\delta^{13}C_{carb}$) were performed on finely ground samples of the palaeolake sections when the carbonate content exceeded 10%. Depending on the carbonate content, 30 to 250 μg of sample material was loaded in vials and measured at the GeoForschungsZentrum Potsdam using a carbonate device (KIEL IV) coupled to an isotopic ratio mass spectrometer (MAT253, ThermoFisher Scientific). Each sample was acidified under vacuum with 103% $H_3PO_4$ at 72 °C for 10 min, and the released $CO_2$ was cryogenic trapped and purified for highly sensitive measurements. The oxygen and carbon isotopic composition are reported using the conventional delta notation (δ) in ‰ relative to VPDB (Vienna PeeDee Belemnite) calibrated with the international standards (NBS 19 and NBS 18) and one internal laboratory standard (C1). The standard deviation (1 σ) for the reference samples is lower than 0.08‰ for both $\delta^{18}O$ and $\delta^{13}C$.

## Regional climate simulations

Many palaeoclimate simulations are carried out with models of coarse horizontal resolutions of several hundreds of kilometres. In these simulations, sub-grid scale processes such as deep convection need to be parameterized, resulting in a poor simulation of precipitation regarding the diurnal cycle and the type of precipitation[49–51]. Most importantly for our present study, the effect of steep and high orography on local atmospheric circulation cannot be represented. Therefore, we use the first storm-resolving simulations of mid-Holocene North Africa by Jungandreas et al.[26,51]. These simulations were performed by using the ICON-NWP (ICOsahedral Nonhydrostatic model framework for numerical weather prediction) version 2.5.0[52], which is also used as a standard model framework by the German Weather Forecasting Service. The storm-resolving simulations were embedded in a set of simulations with different spatial resolution: regional climate simulations on a larger regional domain covering Africa and the tropical and mid-latitude Atlantic and Indian ocean at a spatial resolution of 40 km were forced by initial and lateral boundary conditions taken from a transient global Holocene simulation of the Max Planck Institute Earth System Model (MPI-ESM)[53]. The simulations at 40 km resolution were run over 30 years, including 15 years for spin-up and 15 years for analysis. (The first year for the analysis, labelled year 1 in the Supplementary Fig. 6 and in the Supplementary Tables 2, 3, is calculated in the model using the parameter of the Earth's orbit at 6974 y BP.). From these 15 years, a year of a strong monsoon (labelled year 3) and of a weak monsoon (labelled year 4) were chosen. Data from these two years were used to drive simulations at a resolution of 20 km, then at 10 km, and finally at the storm-resolving 5 km on smaller domains. Land surface parameters were taken from present-day reanalysis data of the Integrated Forecast System (IFS) of the European Centre for Medium-Range Weather Forecasts (ECMWF) and were adjusted in order to represent denser, mid-Holocene-like vegetation cover. No dynamic lake model is implemented, hence no Tibesti palaeolakes exist in the regional model (for further details see Jungandreas et al.[26]).

We analyse the set of simulations with different horizontal resolutions in the following way. First, we use the simulation with a horizontal resolution of 40 km and compute the annual precipitation cycle averaged over the entire Tibesti domain (Figure. S6). The Tibesti Mountains receive $252 +/- 101$ mm during the months of July, August and September (JAS) on average over the 15 years. During the strong monsoon year, the precipitation sum for JAS amounts to some 390 mm, and during the weak monsoon year, to some 70 mm. Within the 15 years of the analysis period, we find two years of similar strong and weak monsoonal precipitation, showing that the selected years of strong and weak monsoons are not outliers. We focus on the strong monsoon case to highlight the physical processes that lead to the strong rainfall over the Tibesti. To upscale the precipitation in the 5 km experiments from seasonal (June – October) means to annual means, the ratio of summer-monsoonal to annual precipitation derived from the 40 km-simulation has been used, assuming that the seasonality is not affected by the horizontal resolution. An assessment of the modelling results is given in the Supplementary Notes (Supplementary Fig. 7).

## Palaeohydrology

For studies on the past water balance of lakes, the availability of hydrological data is usually limited and insufficient for the use of complex hydrological models. Therefore, we take the general water balance equation which has frequently been applied to study basic hydrological conditions of recent lakes and palaeolakes in data-scarce regions in Africa or other regions worldwide[54–62]. The general water balance for endorheic lakes, i.e., lakes without outflow, states that

$$\Delta S = (P - E_A)A_L + R_C A_C + G_D A_D - G_L A_B \qquad (1)$$

where $\Delta S$ is the change of the water volume of a lake, $P$ the precipitation on the lake surface area $A_L$ and $E_A$ the actual evaporation from the lake surface. $R_C$ is surface runoff into the lake from a catchment area $A_C$ around the lake. $G_D$ refers to the water that may percolate from the endorheic part of the caldera systems with an area $A_D$, in our case the Yirrigué caldera in which the crater Trou au Natron is embedded and the Emi Koussi summit caldera in which the Era Kohor is embedded. Groundwater may also leak from the lake into the ground below the lake, indicated by $G_L$ and the bottom area $A_B$. $\Delta S$ is given as water volume, and the area-specific variables $P$, $E_A$, $R_C$, $G_D$, $G_L$ as water column.

Detailed remote sensing and GIS-based topographical analysis was used to estimate the surface drainage area $A_D$ of the Yirrigué caldera and Emi Koussi summit caldera that could feed the Trou au Natron and the Era Kohor, respectively (Fig. 1). The internal drainage systems were mapped and lake areas estimated. The analysis was performed in ArcGIS using the digital elevation models TANDEM-X at Era Kohor (spatial resolution 12 m) and ASTER GDEM at Trou au Natron (spatial resolution of 30 m). This analysis yields estimates for the palaeolake area $A_L$ of 31.6 km² for the Trou au Natron and 3.1 km² for the Era Kohor. The surface drainage areas $A_C$ are found to be 17 km² for the Trou au Natron and 4.4 km² for the Era Kohor.

In a long-term equilibrium, $\Delta S$ is equal to 0, thus

$$0 = (P - E_A) + R_C A_C/A_L + G_D A_D/A_L - G_L A_B/A_L \qquad (2)$$

Actual evaporation $E_A$ of open water bodies is assumed to be equal to the potential evaporation PE. Because detailed information on the wind flow and the energy budget of the crater palaeolakes is not available, we use the classical approach by Budyko[28] as a first approximation. According to this, $PE = R_{net}/l_v$ where $R_{net}$ is the net radiation budget, i.e. the sum of incoming solar radiation, reflected solar radiation, absorbed atmospheric thermal radiation and emitted thermal radiation. We also assume that the albedo (reflectivity of solar radiation) and the emissivity (of thermal radiation) of the lake surface is close to the values of the surrounding area, in our case, the Tibesti volcanic rock.

A general challenge of lake water balance calculations is a lack of knowledge on groundwater fluxes (input and loss) even for recent lakes and especially for palaeolakes. Therefore, we derive the hydrological balance to yield the net groundwater flux $G = G_D A_D/A_L + G_L A_B/A_L$ for an assessment of whether a stable lake existed[58–60]. The final equation for the crater lakes of the Trou au Natron and the Era Kohor is thus:

$$0 = P - PE + R_C A_C/A_L + G \qquad (3)$$

Precipitation and potential evaporation are taken from the ICON-NWP simulations and upscaled from seasonal to annual mean values (see above). The annual mean values for the strong and the weak monsoon years are extrapolated to multi-year mean values by assuming a linear relation

$$P_5 = a + bP_{40} \qquad (4)$$

with $a = (P_{40}{}^S P_5{}^W - P_{40}{}^W P_5{}^S)/(P_{40}{}^S - P_{40}{}^W)$ and $b = (P_{40}{}^S P_5{}^W - P_{40}{}^W P_5{}^S)/(P_{40}{}^S - P_{40}{}^W)$, where the subscripts 5, 40 refer to the climate simulation at 5 km and 40 km resolution, respectively, and the superscripts indicate the strong (s) and weak (w) monsoon years.

The inflow volume from the surface catchments of the lakes is calculated from precipitation using a runoff coefficient of r = 0.59, following the values for bare loamy sands with a slope >10% in Mahmoud and Alazba[52]. Assuming that the precipitation rate over the lake is as large as over the catchment area and that the evaporation of water flowing from the catchment into the lake is negligibly small, we have

$$R = R_C A_C/A_L = r P A_C/A_L \qquad (5)$$

The values $P$, $PE$, $R$, $G$ and, for comparison with Yacoub et al.[20], the ratio $PE/(P + R)$ are listed in the Supplementary Table 2).

## Data availability

The regional climate simulations published in Jungandreas et al.[26] are stored at the World Data Center for Climate (WDCC), hosted and maintained by the German Climate Computing Center (DKRZ) in Hamburg, Germany: https://www.wdc-climate.de/ui/entry?acronym=ICON-NWP_mH_pd. Sedimentary data and model output data to reproduce numbers and figures in the article are archived and are publicly available at https://doi.org/10.17617/3.T49VMF (Claussen, 2025).

## Code availability

The code and further information on the regional climate simulations are described in Jungandreas et al.[26] (https://cp.copernicus.org/articles/19/637/2023) and available as supplement to the publication: http://hdl.handle.net/21.11116/0000-000A-E8FB-6. Analysis and plot scripts that have been used in this study are archived and are publicly available at https://doi.org/10.17617/3.T49VMF (Claussen, 2025).

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

## Acknowledgements
This work contributes to the project PalMod, funded by the German Federal Ministry of Research, Technology and Space (BMFTR), Research for Sustainability initiative (FONA, http://www.fona.de, last access: 3rd February 2022). A.D. was financed by PalMod (grant no. 01LP2306A). M.C. acknowledges funding of an emeritus working group by the Max Planck Society. M.D. is financed by the DFG (Deutsche Forschungsgemeinschaft) within the Priority Programme SPP2143 "Entangled Africa" (project number 404354295). L.J. was funded by funded by the Saxon State Ministry for Science, Culture and Tourism (SMWK) – [3-7304/44/4-2023/8846]. S.K. acknowledges support of the DFG within Collaborative Research Centre "Our Way to Europe" (project 57444011-SFB806) for fieldwork in the Tibesti in 2015 and 2016 and credits Baba Mallaye, former director of Centre National d'Appui à la Recherche (CNAR), N'Djaména, for his crucial assistance in Chad. The authors thank the Integrated Climate Data Center of Center for Earth System Research and Sustainability (CEN), Universität Hamburg, for providing the TAMSAT data. Open access funding is provided by Freie Universität Berlin, Germany.

## Author contributions
P.H. and M.C. contributed equally to this work. P.H., M.D. and F.D. provided and analysed the sedimentological and geochemical data. B.S. analysed the stable isotope composition on carbonates. M.C., A.D. and L.J. analysed the model simulations. C.R.-I. provided the hydrological analysis. S.K. initiated the recent scientific exploration of the Tibesti crater lakes and headed the fieldwork. S.K. and F.D. realized the sampling in the field. P.H. and M.C. designed the study and wrote a first draft of the paper and all authors contributed to the discussion and the revision of the paper.

## Funding

## Competing interests
The authors declare no competing interests.
