## [Transparent Peer Review file · Nature Communications]

Mid-Holocene extreme precipitation in the Tibesti (Central Sahara)

Corresponding Author: Dr Philipp Hoelzmann

Version 0:

Reviewer comments:

Reviewer #1

(Remarks to the Author)
Summary

The authors use very high resolution (5 km) climate simulations to show how substantial orographic rainfall may have occurred over the East side of the Tibesti mountains in the Sahara during the early to mid-Holocene. This is found to arise due to the interaction of enhanced monsoonal flow, moisture sourced from the Mediterranean and orographic uplift of air masses. The results are convincing and I have only minor comments.

I restrict my comments to the modelling. Overall, the results are convincing and provide valuable insight into the conditions that generated climate signals at these sites. I have one main query: the authors argue that the moisture is sourced from the Mediterranean but is converted to precipitation by interaction with the shallow monsoonal flow and orography. Using your suite of runs can you say whether this response weakens in the progressively lower resolution Holocene results (e.g. at 10, 20 or 40 km resolution)? I realise with limited realisations of different seasons it may be difficult to show this clearly but it would support your central argument on the modelling side of this work. It would also be interesting to know whether a 40~km simulation is good enough or whether it too would underestimate the total rainfall? This analysis could possibly also show whether the results are converging at 5~km or whether even higher resolution could be expected to further increase the rainfall response?

Minor comments

Figure 2:

The arrows on the wetter-drier lines are not quite right for the wetter end - they seem to point rightwards but should point leftwards if I understand the figure correctly? Please could you clarify?

Lines 137: "as shown by significantly low $\delta^{18}\text{O}_{\text{carb}}$ values of -6‰ at 7.2 cal ky BP" these values look more like -3‰ to me and 6 is outside of the range of the figure. Could you clarify please?

Line 159: "When extrapolating the seasonal values to the entire year (Methods),"

This sentence could imply a more complex approach than used. Could you instead say something like:

"We extrapolated the seasonal values to the entire year by comparison with the monthly climatology simulated at coarser resolution"

Figure 4. This caption could be clarified e.g. please edit to avoid brackets within brackets where possible. Also could these locations or transects be indicated on figure 3 to clarify the locations shown here.

Generally, in some places the labelling is incorrect (e.g. extended data figure should be supplementary data figure etc).

Supporting Information

Line 145-149: "To this end, we choose a reference site (grid point centre: 11°E, 13°N) in the 145 reconstructions by 19Bartlein et al., 2011 and the corresponding grid cell in the model. This location is 146 chosen because out of the few proxy sites, this site and the corresponding grid cell are located 147 roughly in the centre of the main monsoon rain belt and

because the simulated precipitation sum 148 over the grid box considered approximately equals the reconstructed precipitation sum.”

I'm not sure which analysis this is referring to. Is it shown somewhere?

Line 162-163: “The outstanding high rainfall rates in the Tibesti Mountains compared to the surrounding are also 162 observed during present-day in years with strong monsoon activity (Supplementary information 163 Fig.1). “
Is it the correct figure number?

Reviewer #2

(Remarks to the Author)

A refined knowledge of the African Holocene humid period is of broad interest to the public because it also touches questions of how subtropical dry regions respond to future warmer conditions. The authors combine new proxy data from dried lakes in the high elevation Tibesti region in the central Sahara with climate modelling over north Africa. This combination is highly welcome. The proxy results show a surprisingly strong hydrologic differences between two nearby lakes. This apparent discrepancy is well explained by climate modelling which do not only demonstrate the strong topographic effect leading of local uplift of air masses and rain, but also shifts in moisture sources during the Holocene. Hence, the authors link successfully proxy records with detailed climate modelling. Basically, modelling yield two stages: the early Holocene with a moisture source from southeast by the East African monsoon, and the mid Holocene with a moisture source from the Mediterranean in the northeast. The authors may think a bit more about their presented high-resolution data and how they show variability patterns which might be characteristic to these two stages. This may add further evidence to their conclusions. The manuscript is well written and the methods sound. I made some comments where the text and figures should be improved.

Reviewer #3

(Remarks to the Author)

The study presents a combination of sedimentary data, high-resolution climate modeling, and water balance modeling to estimate the mid-Holocene water balance of two crater lakes in the Tibesti Mountains. The emphasis on orographic precipitation as a major hydrological factor is compelling and challenges previous assumptions about the dominance of the West African Monsoon. The study is particularly interesting because investigating the role of orographic precipitation based on geologic evidence requires the presence of depositional landforms in mountainous regions. Typically, mountains are primarily erosional environments where water and wind rapidly erode climatic and environmental archives. However, the unique aspect of this study is the presence of craters, which provided sufficient accommodation space to trap and preserve these climatic records.

The paper is clear and well-written. I recommend publication after addressing the comments below.

1. Lines 77-87. I feel that the first paragraph in the Results section reads more like a Methods description, as it does not contribute much to the results but rather explains the procedures used by the authors. It would be better if the authors either shorten it to highlight the objective or move it to the Methods section.

2. Lines 111-115. “By 8.2 cal ky BP, calcite precipitation and increased $\delta^{18}\text{O}_{\text{carb}}$ values point to lake level lowering and drier conditions (III: 8.3 – 7.9 ky cal BP), probably related to the 8.2 ka Northern Hemisphere cooling event (32Alley et al., 1997). The highest lake levels (phases IV and V) occurred between 7.9 and 6.6 cal ky BP when sedimentation began also in the higher section W99.”

This is a really interesting point, and it's important that we continue to observe this event imprinted in the geologic record in the Sahara Desert. I was wondering if the authors could compare the duration of this event to non-mountainous settings to highlight any differences. The accommodation space provided by these crater lakes would likely preserve a more complete archive compared to those with less accommodation space, which have been subject to erosion over the past few thousand years.

For example, based on the reported ages, this event appears to have persisted for nearly 400 years, which is surprising—more than twice the duration recorded in the Greenland ice cores (~160 years; Thomas et al., 2007). Additionally, the duration observed in this study is almost three times longer than that recorded in nearby Lake Yoa, where the event lasted only 126 years (Sylvester et al., 2024).

Two possible explanations for this discrepancy are:

1. The mountainous setting preserved sediments from erosion, as wind erosion is less effective in these environments. Also, the craters provided the necessary accommodation space to capture the entire record of the 8.2 ka event.
2. There is a hydrologic mechanism related to monsoon dynamics in these mountain settings that the authors need highlight. I recommend that the authors reflect on this comment and further explore these explanations.

Thomas, E. R., Wolff, E. W., Mulvaney, R., Steffensen, J. P., Johnsen, S. J., Arrowsmith, C., White, J. W. C., Vaughn, B., & Popp, T. (2007). The 8.2ka event from Greenland Ice cores. *Quaternary Science Reviews*, 26(1–2), 70–81.

<https://doi.org/10.1016/j.quascirev.2006.07.017>

Sylvestre, F., Melles, M., Wennrich, V., Dinies, M., Chalié, F., Swingedouw, D., Dallmeyer, A., Shi, X., Claussen, M., Jaeschke, A., Cocquyt, C., Karls, J., Kuper, J., Mallaye, B., Mazur, J.-C., Pailles, C., Rirongarti, R., Rethemeyer, J., Ritter, B., ... Kröpelin, S. (2024). Drought Events during the African Humid Period Suggest Future Drying of the Sahara.

<https://doi.org/10.21203/rs.3.rs-5288623/v1>

3. Lines 122-134. In the second W566 record from the Era Kohor crater, the data capture humidity from 9.5 to 5.4 cal kyr BP, with a highstand lake occurring between 7.9 and 6.5 ka. I was wondering whether you also observe any influence of the 8.2 ka event. If not, could you provide an explanation? Given the similar geological and largely comparable geographic setting, why is the event not recorded here? Could it be due to the smaller water volume in this section, which is only 4.3% of the other lake, leading to lower sediment accumulation that failed to capture the event?

4. Line 174-177. "It may seem surprising that the Mediterranean Sea is the main moisture source for precipitation over the Tibesti, as it was previously assumed that a stronger southerly West African monsoon could have been the main factor (e.g. Yacoub et al., 2023)."

It is surprising that the influence of the Mediterranean reached as far as the Tibesti Mountains. Is there a way for models to provide a quantitative estimate of the contribution from each moisture source? I think this could be important for understanding the potential extent and magnitude of each source's impact on Saharan landscapes.

5. One minor comment. It would be helpful to indicate the locations of the lake cores in Figure 1.

I hope these comments are helpful.

Sincerely,
Abdallah Zaki

Version 1:

Reviewer comments:

Reviewer #1

(Remarks to the Author)

I am happy to say that the authors have addressed all of my comments on the original submission in a satisfactory manner.

Reviewer #2

(Remarks to the Author)

Thank you for addressing my comments thoroughly. The raised points became much clearer now and improved the paper.

Reviewer #3

(Remarks to the Author)

I have just carefully reread the revised manuscript. After reviewing the authors' responses to the reviewers' comments, I am very pleased to see that they have comprehensively addressed the points I raised previously, as well as those made by the other reviewers. As a result, the manuscript is significantly improved. I therefore recommend publication.

Best wishes,
Abdallah Zaki

**REVIEWER COMMENTS and AUTHORS' REPLY**

We would like to thank the editor and the reviewers for their effort and, particularly, for their
constructive comments which help to improve our manuscript.

In the following we reply to the reviewers' comments point by point. The reply is written using blue
letters.

**Reviewer #1 (Remarks to the Author) (Reply by the authors):**

Summary

The authors use very high resolution (5 km) climate simulations to show how substantial orographic
rainfall may have occurred over the East side of the Tibesti mountains in the Sahara during the early
to mid-Holocene. This is found to arise due to the interaction of enhanced monsoonal flow, moisture
sourced from the Mediterranean and orographic uplift of air masses. The results are convincing and I
have only minor comments.

**Reply:** We thank Reviewer #1 for insightful comments.

I restrict my comments to the modelling. Overall, the results are convincing and provide valuable
insight into the conditions that generated climate signals at these sites. I have one main query: the
authors argue that the moisture is sourced from the Mediterranean but is converted to precipitation
by interaction with the shallow monsoonal flow and orography. Using your suite of runs can you say
whether this response weakens in the progressively lower resolution Holocene results (e.g. at 10, 20
or 40 km resolution)? I realise with limited realisations of different seasons it may be difficult to show
this clearly but it would support your central argument on the modelling side of this work. It would
also be interesting to know whether a 40~km simulation is good enough or whether it too would
underestimate the total rainfall? This analysis could possibly also show whether the results are
converging at 5~km or whether even higher resolution could be expected to further increase the
rainfall response?

**Reply:** We very much like this suggestion. Therefore, we modified Fig. 3 to show the differences in
the precipitation and potential evaporation patterns in the Tibesti region between the 5km
simulation and the 40km simulation. We also put the orography plot, the former Supplementary
Fig.4, into the main text to highlight the strong smoothing of orography in the 40 km simulation – not
to speak of the resolution in a global model which runs at a resolution of approximately 200 km. It
becomes evident that the 40km simulation hardly captures the orographic effect of the steep Tibesti
on the precipitation. In fact, the atmospheric hydrological balance is highly negative. Hence, the
Tibesti paleo lakes could not exist in the 40km simulation without a strong ground water flux into
both craters. We also modified the text to better explain and highlight the results. And we added the
hydrological estimate for the 40 km simulation to Tab. S2 and Tab. S3 (values in brackets). Finally, we
harmonized Fig. S6 and Tab. S2, S3 by presenting the last 15 years, instead of the last 17 years, of the
simulation in Tab. S2, S3.

In order to judge whether the results are converging at some 5 km resolution in which still many
orographic details of the craters are not realistically represented, we would need to perform
additional simulations at a finer grid. Such simulations would be extremely costly, because the entire
modeling system would have to be implemented on the new HPC system which in turn would require

considerable additional person power. This cannot be done in the frame of the present paper. It
would, however, be an interesting topic for an additional study.

**Minor comments**

Figure 2:

The arrows on the wetter-drier lines are not quite right for the wetter end - they seem to point
rightwards but should point leftwards if I understand the figure correctly? Please could you clarify?

**Reply:** We thank Reviewer #1 for addressing this issue. We changed the directions of the arrows on
the wetter end to point leftwards.

Lines 137: "as shown by significantly low $\delta^{18}\text{O}_{\text{carb}}$ values of -6‰ at 7.2 cal ky BP" these values look
more like -3‰ to me and 6 is outside of the range of the figure. Could you clarify please?

**Reply:** We thank Reviewer #1 for indicating this mismatch. We rescaled the axis in Fig. 2 accordingly.

Line 159: "When extrapolating the seasonal values to the entire year (Methods),"

This sentence could imply a more complex approach than used. Could you instead say something
like:

"We extrapolated the seasonal values to the entire year by comparison with the monthly climatology
simulated at coarser resolution"

**Reply:** We realize that this sentence could be misleading. Therefore, we have shortened this
sentence, because the upscaling from seasonal to annual values is described in detail in the Methods.

Figure 4. This caption could be clarified e.g. please edit to avoid brackets within brackets where
possible. Also could these locations or transects be indicated on figure 3 to clarify the locations
shown here.

**Reply:** We like the suggestion and remove the double brackets. We have checked the possibility of
indicating the locations of the cross sections shown in Fig. 4 in the Fig. 3. The Fig. 3 becomes pretty
messy with more lines in the plots. On the other hand, we are convinced that the location of the
craters and, hence, the location of cross section are clearly identifiable from the P and the P-PE
maxima shown in the Fig.3 b and 3 c. and from the orography plot, now Fig. 5. We also add the
location of the craters to the caption of Fig. 3.

Generally, in some places the labelling is incorrect (e.g. extended data figure should be
supplementary data figure etc).

**Reply:** We thank Reviewer #1 for this indication. We changed the labelling to Fig. S1, Fig. S2 etc. and
Tab. S1, Tab. S2 etc. accordingly.

**Supporting Information**

Line 145-149: "To this end, we choose a reference site (grid point centre: 11°E, 13°N) in the 145
reconstructions by 19Bartlein et al., 2011 and the corresponding grid cell in the model. This location
is 146 chosen because out of the few proxy sites, this site and the corresponding grid cell are located

147 roughly in the centre of the main monsoon rain belt and because the simulated precipitation
sum 148 over the grid box considered approximately equals the reconstructed precipitation sum.”

I’m not sure which analysis this is referring to. Is it shown somewhere?

**Reply:** The text “Assessment of climate modelling results” was misleading because of wrong
references to figures. We changed the text pieces. Basically, the upper two figures (Fig. S7 a, b) are
the differences in precipitation between the reference site (grid point centre: 11°E, 13°N) and all
other grid boxes from the mid Holocene 5km simulations of the strong and weak monsoon year. The
middle row figures (Fig. S7c, d) are the precipitation values of these 5k simulations for the strong and
weak monsoon year. The lower figure (Fig. S7 e) is the precipitation derived from satellite data for a
*present-day* strong monsoon year.

Line 162-163: “Mountains compared to the surrounding are also 162 observed during present-day in
100 years with strong monsoon activity (Supplementary information 163 Fig.1). “

Is it the correct figure number?

**Reply:** We thank Reviewer #1 for careful reading. Indeed, the figure number in the text was wrong. It
should read “Fig. S7e”.

**Reviewer #2 (Remarks to the Author) (Reply by the authors):**

A refined knowledge of the African Holocene humid period is of broad interest to the public because
it also touches questions of how subtropical dry regions respond to future warmer conditions. The
authors combine new proxy data from dried lakes in the high elevation Tibesti region in the central
Sahara with climate modelling over north Africa. This combination is highly welcome. The proxy
results show a surprisingly strong hydrologic differences between two nearby lakes. This apparent
discrepancy is well explained by climate modelling which do not only demonstrate the strong
topographic effect leading of local uplift of air masses and rain, but also shifts in moisture sources
during the Holocene. Hence, the authors link successfully proxy records with detailed climate
modelling. Basically, modelling yield two stages: the early Holocene with a moisture source from
southeast by the East African monsoon, and the mid Holocene with a moisture source from the
Mediterranean in the northeast. The authors may think a bit more about their presented high-
resolution data and how they show variability patterns which might be characteristic to these two
stages. This may add further evidence to their conclusions. The manuscript is well written and the
methods sound. I made some comments where the text and figures should be improved.

**Reply:** We would like to thank Reviewer #2 for constructive critique and comments which we will
respond to in more detail below.

**Main Text:**

Page 5, Line 67: What was the outcome of this study and which questions remained open and can be
answered by high-resolution analysis? Is this necessary for climate modelling – the main task of the
paper – which rather uses general climatic conditions? How can the high-resolution results support
the modelling results? In the modelling part high-resolution results are not addressed.

**Reply:** We thank Reviewer #1 for this question and giving us the opportunity to point out more
clearly why the high-resolution sediment analysis was useful for our study. Yacoub et al. (2023) relate
their data from the two crater palaeolake records rather generally to the African Holocene Humid
Period (AHHP) without differentiating between the Trou au Natron and the Era Kohor: “..., our data
support the notion that the timing of the Tibesti palaeolakes` most humid conditions and associated
highstand lakes are broadly synchronous with the peak of the AHHP reached in the interval between
~9500 and ~ 8000 cal yr BP, which is commonly expressed as a global well spell over the Sahara and
Sahel regions...” (Yacoub et al. 2023).

With our higher resolved analysis we could address the lake evolution for both locations with more
detail:

(1) We show that at least the record in the Trou au Natron (JK48) is climatically significant as we see a
signal for a short-term lake level lowering possibly related to the 8.2 ka event;

(2) we could identify and differentiate varying lakes phase and we prove that the period around c. 7.0
141 cal ky BP was the most stable and probably deepest lake phases for both lakes (Fig. 2: lake phase V in
the Trou au Natron; lake phase b in the Era Kohor). This period is represented in the spatially highly
resolved climate simulation.

(3) we identified differences in the timing of the evolution of the lakes. The section W566 in the Era
Kohor summit caldera of the Emi Koussi represents a littoral position of a lake level highstand – 130
146 m above the present crater floor – as early as 9.5 cal ka BP. Within the Trou au Natron section JK48 at
147 a profundal location represents for this period an initial and instable lake phase as represented by
148 the Mn/Fe and (Ca/(Fe + Al + Si) sedimentary data. Sedimentation at section W99 at a littoral
position within the Trou au Natron – 330 m above section JK48 – only sets in later at around 8000 cal

a BP. This point is picked up in the interpretation of the simulated climate data and in the conclusions
when we infer that the Trou au Natron on the east-west orientated branch of the Tibesti should have
benefited more from the uplift of the mid-Holocene northerly flow, and the Era Kohor on the
southern edge of the north-south orientated branch (see Fig. 3), more from the intense early
Holocene southwesterly monsoon flow. Because of the extreme HPC requirements, we could make
use of only one set of high-resolution regional climate simulations for mid Holocene climate
conditions. For exploring possible differences between early and mid Holocene monsoon flow in
Africa, we had to consider published global palaeoclimate simulations.

To address the suggestion of Reviewer #2, we added a number of sentences and words in lines 56-
60; line 74; line 292, and line 310.

Page 6, Line 81: You mean more detrital input during dry periods which goes hand in hand with
carbonate precipitation? Carbonate precipitation is not directly reflected in Ti and AL

**Reply:** We thank Reviewer #2 for his comment which shows, that our spelling of the parameter
representing authigenic carbonate precipitation caused confusion. We changed this in the text from
"...analyses of Ca/ Σ Fe, Ti and Al relate ..." to: "... analyses of Ca/(Fe + Ti + Al) relate ...". This term of
the parameter is also used in Figure 2.

Page 6, Line 82: bulk?

**Reply:** We thank Reviewer #2 for this question. He/She is right as we did not distinguish between
authigenic and detrital carbonate within the isotope analyses. To make this clear we substituted
"authigenic" by "bulk".

Page 7, Fig. 2: Add headlines (profundal, littoral)

**Reply:** We thank Reviewer #2 for this suggestion and we added the proposed headlines in Figure 2.

Page 7, Fig. 2 Caption: The figure caption contains already descriptions and interpretations which are
also part of the text, hence there is some repetition when reading the text. Check with the journals
guidelines if this is desirable.

**Reply:** We thank Reviewer #2 for this suggestion. In the 'Guide to Authors' it is stated: "*Include*
*enough information to understand the figure without referring to the text.*" Therefore, we keep the
descriptions and interpretations in the caption but we will follow the next point of Reviewer #2 so
that the caption of Figure 2 is shortened.

Page 6, Fig. 2 Caption: I think these two sentences are unnecessary because this becomes clear from
the figure and legend.

**Reply:** We thank Reviewer #2 for this suggestion and we deleted the two last sentences of this figure
caption.

Page 9, Line 150: I would add: “although their potential drainage areas are comparable in size?”

**Reply:** We thank Reviewer #2 for this suggestion. However, this is only roughly true for the surface
catchment (Era Kohor: ca. 4.4 km², Trou au Natron ca. 17 km²). However, if we include for both
systems the lake areas (lake area + surface catchment) the dimensions differ substantially: Era Kohor
7.5 km², TaN: c. 45 km²). From our point of view the TaN is a much bigger system. We will include the
numbers mentioned above in the caption of Figure 1.

Page 12, Fig.4: I suggest to put headlines to the model boxes for easier readability. The figure caption
is rather complex to read and it link to the graphs.

**Reply:** We agree with the Reviewer #2 that headlines make the figure easier to understand. Hence
we added headlines, but we did not, however, add all suggested headlines, because the figure
became too messy.

Page 14, Fig. 255/256: If I understand right this is concluded from the Mn/Fe. Isotopes are missing. Is
this enough proxy evidence to come up with the following hypothesis?

**Reply:** We thank Reviewer #2 for this comment and the possibility to clarify our interpretation.
Isotopes were only possible to measure if the bulk carbonate contents exceeded 1 %. For section
W566 in the Era Kohor crater the carbonate content exceeded 1 % only for the four samples at the
top (from 6.2 cal ka BP onwards). However, we have other arguments to distinguish the general lake
evolutions between the Trou au Natron and the Era Kohor craters:
- the littoral position of section W566 (Era Kohor) was already reached by 9.5 cal ka BP.
- at this time the Trou au Natron exhibits only sediments at a profundal location (JK48)
- the sedimentary geochemical data of Mn/Fe and Ca/(Fe + Al + Si) at this profundal location (JK48)
represents a shallower water depth;
- at the littoral location of the Trou au Natron crater (section of W99) the sedimentation began only
from 8.0 cal ka BP onwards.

With the suggestion of Reviewer #2 to add headlines to the sections (see above: JK48 profundal; W99
littoral; W566 littoral) this has been clarified.

**Supplementary Information**

Page 2, Line 30: You should add coordinates of the sections presented in the paper.

**Reply:** We thank Reviewer #2 for this suggestion and added in the Supplementary Information the
coordinates in the section “Lithology” where the sediment archives are described in detail:
- W566 (19°50.400’N / 18°33.101’E at 2774 m asl)
- JK48 (20°58.349’N / 16°33.157’ E at 1560 m asl)
- W99 (20°57.016’N / 16°33.110’E at 1890 m asl)

Page 5, Line 75: Could you show the depth age models?

**Reply:** We thank Reviewer #2 for this suggestion and we added the age-depth models in the
Supplementary Information within the subheading “Dating and age depth model” with figures S3
(section JK48); S4 (section W99); S5 (section W566).

**Reviewer #3 (Remarks to the Author) (Reply by the Authors)**

The study presents a combination of sedimentary data, high-resolution climate modeling, and water
balance modeling to estimate the mid-Holocene water balance of two crater lakes in the Tibesti
Mountains. The emphasis on orographic precipitation as a major hydrological factor is compelling
and challenges previous assumptions about the dominance of the West African Monsoon. The study
is particularly interesting because investigating the role of orographic precipitation based on geologic
evidence requires the presence of depositional landforms in mountainous regions. Typically,
mountains are primarily erosional environments where water and wind rapidly erode climatic and
environmental archives. However, the unique aspect of this study is the presence of craters, which
provided sufficient accommodation space to trap and preserve these climatic records.

The paper is clear and well-written. I recommend publication after addressing the comments below.

**Reply:** We would like to thank Reviewer #3 for the constructive comments.

1. Lines 77-87. I feel that the first paragraph in the Results section reads more like a Methods
description, as it does not contribute much to the results but rather explains the procedures used by
the authors. It would be better if the authors either shorten it to highlight the objective or move it to
the Methods section.

**Reply:** We thank Reviewer #3 for this suggestion. We follow Reviewer #3 and left only the first
sentence from this paragraph in the main text and moved the remaining paragraph into the Methods
section.

2. Lines 111-115. “By 8.2 cal ky BP, calcite precipitation and increased $\delta^{18}\text{O}_{\text{carb}}$ values point to lake
level lowering and drier conditions (III: 8.3 – 7.9 ky cal BP), probably related to the 8.2 ka Northern
Hemisphere cooling event (32Alley et al., 1997). The highest lake levels (phases IV and V) occurred
between 7.9 and 6.6 cal ky BP when sedimentation began also in the higher section W99.”

This is a really interesting point, and it’s important that we continue to observe this event imprinted
in the geologic record in the Sahara Desert. I was wondering if the authors could compare the
duration of this event to non-mountainous settings to highlight any differences. The accommodation
space provided by these crater lakes would likely preserve a more complete archive compared to
those with less accommodation space, which have been subject to erosion over the past few
thousand years.

For example, based on the reported ages, this event appears to have persisted for nearly 400 years,
which is surprising—more than twice the duration recorded in the Greenland ice cores (~160 years;
Thomas et al., 2007). Additionally, the duration observed in this study is almost three times longer
than that recorded in nearby Lake Yoa, where the event lasted only 126 years (Sylvester et al., 2024).

Two possible explanations for this discrepancy are:

1. The mountainous setting preserved sediments from erosion, as wind erosion is less effective in
these environments. Also, the craters provided the necessary accommodation space to capture the
entire record of the 8.2 ka event.

**Reply:** We thank Reviewer #3 for this very interesting and inspiring comment. However, we are
relatively reluctant with this idea to further interpret the duration of the 8.2ka-event from our
sedimentary records for the following reasons:

- due to the lack of larger charcoal pieces within the sediments of sections JK48 and W566 these age-

depth models were performed from charred plant particles (cpc; Table S1) or uncharred plant
parenchym (upp). These ¹⁴C ages show errors between ±20 to ±130 years (Table S1) and therefore
the precision may not allow for calculations within 100a to 200a;
- we see this phase that probably relies to the 8.2 ka event only in section JK48 as section W99 only
shows sediments that were precipitated after the 8.2 ka event;
- in section W566 the monolith with the lakes sediments representing the period 8.1 to 7.1 cal ka BP
(depth 75 cm to 52 cm) simply did not reach our laboratory intact so that we could only take samples
representing the base, the middle, and the top of this monolith (in Figure 2 this is shown by a dotted
line for the Mn/Fe). (The challenging transport conditions included transport out of the crater with
donkeys and further transport to the cars with camels). We mentioned this unfortunate
circumstance now in the description of the lithology in the Supplementary information;
- also section JK48 is a solitary yardang and thus only one remnant of the former lake floor which has
been largely eroded near the rim of the Trou au Natron crater.

Nevertheless, the hint of a lake-level lowering within this period as in section JK48 may point to the
8.2 ka event and may show that the palaeolake within the Trou au natron did react to this event.
Therefore, we picked up the comment of Reviewer #3 and changed the text to:
Calcite precipitation and increased $\delta^{18}\text{O}_{\text{carb}}$ values point to lake level lowering and drier conditions
during a period (III: 8.3 – 7.9 ky cal BP) when the 8.2 ka Northern Hemisphere cooling event (³²Alley
et al., 1997) also affected North Africa and climatic feedbacks may have even prolonged its duration
in the Sahara (Tierney et al. 2016). However, the age-depth model has not the necessary precision to
reliably fix this phase to the 8.2 ka event or to calculate its duration from section JK48.

2. There is a hydrologic mechanism related to monsoon dynamics in these mountain settings that the
authors need highlight.

I recommend that the authors reflect on this comment and further explore these explanations.
Thomas, E. R., Wolff, E. W., Mulvaney, R., Steffensen, J. P., Johnsen, S. J., Arrowsmith, C., White, J. W.
C., Vaughn, B., & Popp, T. (2007). The 8.2ka event from Greenland Ice cores. *Quaternary Science*
*Reviews*, 26(1–2), 70–81. <https://doi.org/10.1016/j.quascirev.2006.07.017>
Sylvestre, F., Melles, M., Wennrich, V., Dinies, M., Chalié, F., Swingedouw, D., Dallmeyer, A., Shi, X.,
Claussen, M., Jaeschke, A., Cocquyt, C., Karls, J., Kuper, J., Mallaye, B., Mazur, J.-C., Pailles, C.,
Rirongarti, R., Rethemeyer, J., Ritter, B., ... Kröpelin, S. (2024). Drought Events during the African
Humid Period Suggest Future Drying of the Sahara. <https://doi.org/10.21203/rs.3.rs-5288623/v1>

**Reply:** We assume that this comment relates to the 8.2 k event. This event is not in the focus of our
study. Including the 8.2 k event in the modelling would require substantial additional HPC and
personal resources. It would be an interesting topic for a new study.

In addition, we already pointed out above why we are reluctant in emphasizing the interpretation of
the 8.2 ka event.

3. Lines 122-134. In the second W566 record from the Era Kohor crater, the data capture humidity
from 9.5 to 5.4 cal kyr BP, with a highstand lake occurring between 7.9 and 6.5 ka. I was wondering
whether you also observe any influence of the 8.2 ka event. If not, could you provide an explanation?
Given the similar geological and largely comparable geographic setting, why is the event not
recorded here? Could it be due to the smaller water volume in this section, which is only 4.3% of the
other lake, leading to lower sediment accumulation that failed to capture the event?

**Reply:** In section W566 the monolith with the lake sediments representing the period 8.1 to 7.1 cal
321 ka BP simply did not reach our laboratory intact so that we could only take samples representing the
322 base, the middle, and the top of this monolith. The challenging transport conditions included
transport out of the crater with donkeys and further transport to the cars with camels.

4. Line 174-177. “It may seem surprising that the Mediterranean Sea is the main moisture source for
precipitation over the Tibesti, as it was previously assumed that a stronger southerly West African
monsoon could have been the main factor (e.g. Yacoub et al., 2023).”

It is surprising that the influence of the Mediterranean reached as far as the Tibesti Mountains. Is
there a way for models to provide a quantitative estimate of the contribution from each moisture
source? I think this could be important for understanding the potential extent and magnitude of each
source's impact on Saharan landscapes.

**Reply:** We like the idea. However, estimating the relative contributions of moisture from the
Mediterranean versus moisture from the West African monsoon would require completely new
simulations using Lagrangian transport modeling. This cannot be done for this paper, but it would be
an interesting idea for a new study.

Actually, we were not surprised to find a strong influence of the Mediterranean Sea on the mid-
Holocene rainfall in the Tibesti mountain complex. Although not directly comparable, it has been
shown that moisture from the Mediterranean Sea becomes the main source of Sahelian rainfall in a
warming climate (e.g. Park et al., Nat.Clim.Change, 2016)

5. One minor comment. It would be helpful to indicate the locations of the lake cores in Figure 1.

**Reply:** Thank you very much for this useful suggestion that we have implemented in Figure 1.